# Erodibility of Nanocomposite-Improved Unsaturated Soil Using Genetic Programming, Artificial Neural Networks, and Evolutionary Polynomial Regression Techniques

Kennedy C. Onyelowe [1,2,*], Ahmed M. Ebid [3], Uchenna Egwu [4], Michael E. Onyia [5], Hyginus N. Onah [5], Light I. Nwobia [2], Izuchukwu Onwughara [6] and Ali Akbar Firoozi [7]

1   Department of Civil and Mechanical Engineering, Kampala International University, Kampala P.O. Box 20000, Uganda
2   Department of Civil Engineering, Michael Okpara University of Agriculture, Umudike 440101, Nigeria; nwobia.light@mouau.edu.ng
3   Department of Structural Engineering, Faculty of Engineering and Technology, Future University, New Cairo 11845, Egypt; ahmed.abdelkhaleq@fue.edu.eg
4   Department of Civil Engineering, School of Civil Engineering and Geosciences, Newcastle University, Newcastle upon Tyne NE1 7RU, UK; uche_egwu@yahoo.com
5   Department of Civil Engineering, Faculty of Engineering, University of Nigeria, Nsukka 410001, Nigeria; michael.onyia@unn.edu.ng (M.E.O.); hyginus.onah@unn.edu.ng (H.N.O.)
6   Nigeria Erosion and Watershed Management Project, Abia State Ministry of Environment and Ministry of Works, Umuahia 440001, Nigeria; izuuonwughara@gmail.com
7   Department of Civil Engineering, University of Botswana, Gaborone 0061, Botswana; firoozia@ub.ac.bw
*   Correspondence: kennedychibuzor@kiu.ac.ug or konyelowe@gmail.com or konyelowe@mouau.edu.ng

**Abstract:** Genetic programming (GP) of four levels of complexity, including artificial neural networks of the hyper-tanh activation function (ANN-Hyper-Tanh), artificial neural networks of the sigmoid activation function (ANN-Sigmoid), evolutionary polynomial regression (optimized with genetic algorithm) (EPR), and intelligent techniques have been used to predict the erodibility of lateritic soil collected from an erosion site and treated with hybrid cement. Southeastern Nigeria and specifically Abia State is being destroyed by gully erosion, the solution of which demands continuous laboratory examinations to determine the parameters needed to design sustainable solutions. Furthermore, complicated equipment setups are required to achieve reliable results. To overcome constant laboratory works and equipment needs, intelligent prediction becomes necessary. This present research work adopted four different metaheuristic techniques to predict the erodibility of the soil; classified as A-7-6, weak, unsaturated, highly plastic, high swelling and high clay content treated with HC utilized in the proportions of 0.1–12% at the rate of 0.1%. The results of the geotechnics aspect of the work shows that the HC, which is a cementitious composite formulated from blending nanotextured quarry fines (NQF) and hydrated lime activated nanotextured rice husk ash (HANRHA), improves the erodibility of the treated soil substantially and consistently. The outcome of the prediction models shows that EPR with SSE of 1.6% and $R^2$ of 0.996 outclassed the other techniques, though all four techniques showed their robustness and ability to predict the target (Er) with high performance accuracy.

**Keywords:** unsaturated soil; ANN-HyperTanh; ANN-Sigmoid; evolutionary polynomial regression (EPR); HC-treated soil; erodibility of pavement materials

## 1. Introduction

### 1.1. Preamble

Soil erosion, being an ancient threatening phenomenon to the productivity of soil, survives or suffers its cause due to soil profile and horizonation, terrain or architecture, soil management, and climate conditions [1,2]. The action of wind and rain instigates failures within the soil profile leading to its slope failure and overall stability degradation [3].

Poesen [4], Onyelowe et al. [1], and Pal et al. [5] described gully erosion as an erosion deep stream feature formed by running water with a cross-sectional area greater than 1 ft$^2$, which is too wide to be damaged by traditional tillage, and mostly occurs in lateritic soils and comparatively in weak rocks of weathered materials. Crop damage due to sand splay, inordinate topographical distortion, lack of reservoir storage capacity, land infertility leading to food scarcity and damage of roads and other infrastructure are further identified as negative consequences of gully erosion [6]. Rainfall-induced slope failures are fundamentally caused by increased pore pressure and seepage forces during periods of severe and prolonged rainfall. The overall result of this process triggers a breakdown and dispersion of the soil aggregates [7,8]. However, prior to this breakdown and dispersion process, the effective stress in the soil will be lowered due to the increased pore pressure and thus reduce soil shear strength, eventually resulting in slope failure [8]. Basically, lighter aggregate materials such as very fine sand, silt, clay and organic matter are eroded earlier than coarser soil aggregates by the raindrop splash and runoff water [7]. Meanwhile, surface water runoff occurs whenever there is excess water on a slope exceeding the degree of saturation of the soil matrix [1]. Moreover, during rainfall a wetting front goes deeper into the slope, resulting in a gradual increase of water content and a decrease of the negative pore water pressure, a phenomenon generally referred to as matric suction [2]. Based on the foregoing erodibility inducement, the loss of suction causes a decrease in the shear strength of the soil on the potential failure surface and finally triggers the failures. Reduced infiltration due to soil compaction, crusting or freezing promotes the surface runoff and soil erosion. Consequently, runoff from agricultural land is more pronounced when compared with other land areas. Fundamentally, the integral properties of the soil such as texture, structure, soil organic matter content, clay minerals, exchangeable cations and water retention and transmission, among others, are critical determining factors of soil erodibility potential [9]. Similarly, there are marked fluctuations of temperature and lowering of the groundwater table leading to the appreciable loss of interstitial moisture of the surface soil matrix propelling the loss of cohesion among the particles aggregating to the soil matrix. The dynamics of this erodibility mechanism are not obviously owing to seasonality effects. Apparently, the washing away of the backfill, subgrade, embankment or surface layer is propelled by seasonal changes which fundamentally result in the weakening of the shear strength of compacted foundation soils, thereby limiting their veracity of usage for foundation or construction purposes [10]. Meanwhile, significant improvement in soil shear strength parameters and soil erosion coefficients can also be obtained by stabilizing the unsaturated soil matrix with non-toxic admixtures [11]. The improvement of erosion coefficients of the soil is mainly attributed to the reduction of the double-layer thickness by the neutralization of surface charges of the soil matrix leading to the formation of more stable particle clusters by polymer bridging [11]. Overall, the interaction among the soil erosion-causing factors and their response to stabilization actions are complex and non-linear, affirming the potential of the use of machine learning techniques to predict them [12,13].

### 1.2. Gully Erosion Disaster in Southeast Nigeria: Efforts of NEWMAP in Mitigating Soil Erosion

Without doubt, various locations have varying degrees of soil erosion they experience owing to the topographic position of slope, vegetation and soil type, and the Southeastern part of Nigeria, particularly Abia State, is apparently amongst the worst hit [1,2,14]. Iro et al. [15] remarked that the loss of soil degrades arable land and fundamentally renders it unproductive. He further maintained that the effects of erosive mechanisms were orchestrated mostly by the recent and rapid increase in the population index in the Southeastern part of Nigeria. Basically, Abia State witnesses a high annual rainfall (about 2000 mm mean/year) with associated high discharge of water as runoff that promotes soil erosion. Moreover, rainfall actions are highly correlated with erosion in all the representative land surface types [14]. This gully erosion problem is affecting sustainable

development because infrastructures such as houses, roads and others are being destroyed by this menace.

Consequently, concerted attempts have been made by various agencies, such as the Nigerian Erosion and Watershed Management Project (NEWMAP)established by Nigeria's ministry of environment to solve the environmental problem faced locally, nationally and globally. For instance, in response to the catastrophic threats of gully erosion and the emerging Land degradation and environmental insecurity, a request for assistance was made in 2010 by the President to the World Bank Nigeria office to support the country in addressing severe erosion and its impacts in south-eastern Nigeria. The Federal Ministry of Environment in commitment with the World Bank and its partner agencies designed the Nigeria Erosion and Watershed Management Project (NEWMAP) to address the menace of gully erosion in the southeast as well as land degradation in the North at a full-blown scale [16]. The project (NEWMAP) is geared towards obtaining Nigeria's Vision 20:2020. The initiative captured a strategic combination of civil engineering, vegetative land management and other catchment protection measures, and community-led adaptive livelihood initiatives. Regrettably, despite these commendable efforts, catastrophic actions of erosion within Abia state have still remained unabated.

### 1.3. Nanostructured Composites for Soil Erodibility Enhancement: Energy and Environmental Sustainability Potential of Using Green Composites and Supplementary Cementitious Materials for Soil Improvement

Basically, the action of soil erosion can lead to the generation of pollution and other associated environmental hazards, especially when green composite materials are not well treated or disposed. Besides the problem of increased pollution and sedimentation in streams and rivers, the clogging of waterways and decline in fish and other species might also result from the lack of proper treatment of various composite materials whose uses in their nanoforms are of immense sustainable value. Moreover, the degraded lands are also often less able to hold onto water, which can worsen flooding. The use of supplementary cementitious materials like nanotextured quarry fine (NQF) and hydrated lime activated nanostructured rice husk ash (HANRHA) blends to improve the soil performance against erosion attacks, and has substantial renewable energy and environmental benefits [17–20]. Meanwhile, there are concerted efforts and policies against continued resource depletion on the local, national and global scales [21,22]. One sustainable way of achieving a carbon-neutral and energy efficient society is by re-using the agro-industrial wastes rather than indiscriminately disposing them into the surroundings, thereby endangering the ecosystem [21]. Soils whose mechanical and physical properties have been improved by incorporating nanomaterials into their matric such as nanocarbons or high-binding green nanocomposites induced into unsaturated lateritic soils could offer a flexible and robust construction material for ecological and environmental sustainability [23–25]. Ucankus et al. [26] remarked that nanomaterials can provide new opportunities for coping with this challenge of environmental remediation as well as treatment of both air and water. Since sustainable geotechnical engineering seeks an effective way to utilize the nanostructured materials such as NQF-HANRHA blends which are predominantly available within the environment to improve soil performance, environmental friendliness and overall food productivity, its application to enhance the erodibility potential of unsaturated soils will be undoubtedly worthwhile.

### 1.4. Evolutionary Computation Techniques for Predicting Soil Erosion and Associated Geotechnical Variables

Erosion models are complex ones because of the uncertainty of data and the dynamics of parameter interactions [27]. Soil erodibility is influenced by surface crust and soil gradation parameters [3,28]. The cover/crop management factor, C factor, which models the effect of vegetation and other land covers, can be predicted or modeled using many methods such as the normalized difference vegetation index (NDVI), which is suitable to estimate the C factor for soil loss assessment with the RUSLE model [29–33], GIS and remote

sensing techniques, and other evolutionary computation methods [34]. Consequently, the use of evolutionary computational techniques for predicting erodibility coefficients appears to be highly promising. Evolutionary computation has been described by a family of algorithms for global optimization inspired by biological evolution and the subfield of artificial intelligence and soft computing [35–37]. In their domain of performance, they possess a metaheuristic or stochastic optimization character as a family of population-based trial and error problem solvers, providing real solution-based options based on the dynamics and nature of the database being analyzed [1,37]. Various evolutionary computation systems have their own variants, architectures and modes of performing their prediction functions.

Yusuf et al. [34] predicted soil erodibility factor, one of the components of the universal soil loss equation, which models the susceptibility of soil to be lost to erosion by employing artificial neural network models using 74 soil series provided by the Department of Agriculture, Malaysia. Puente et al. [38] devised a novel approach for designing new vegetative indices that are better correlated with soil cover factors using field and satellite information. Their approach was based on stating the problem in terms of optimization through genetic programming and building novel indices by iteratively recombining a set of numerical operators and spectral channels until they obtained the most adaptive composite operator.

While some have compared the performance of two or more evolutionary computational techniques [39], others combined the predictive abilities of two or more evolutionary computational techniques in predicting and modelling the behaviours of various engineering systems and parameters, especially as it applies to soil erodibility or resilience responses. Ghorbani et al. [40] predicted the resilient modulus of pavement subgrade soils using the evolutionary polynomial regression (EPR) techniques. Also, Kouchami-Sardoo et al. [28] estimated and predicted soil erodibility to wind using the ERP technique after measuring the wind erosion rate ($gm^{-2} s^{-1}$) at a total of 118 sites in Kerman Province, southeast Iran, with a portable wind tunnel. Their developed EPR model provided a strong basis for the prediction of soil erodibility, where the coefficient of determination ($R^2$) values of 0.89 and 0.87 were obtained by respectively comparing the measured and predicted wind erosion rates for the training and testing data. Arabameri et al. [6] explored the potential of a genetic algorithm-extreme gradient boosting (GE-XGBoost) hybrid computer education solution for spatial mapping of the susceptibility of gully erosion. Their new machine learning approach incorporated the extreme gradient boosting machine (XGBoost) and the genetic algorithm (GA). The GA meta-heuristic they reported was used to enhance the efficiency of the XGBoost classification approach. They further developed a GIS database which contained recorded instances of gully erosion incidents. Their newly built model did not only prove a promising method for large-scale mapping of gully erosion susceptibility, but also demonstrated the performance enhancement that can be obtained by incorporating two or more soft computing techniques. While attempting to implement some remedial measures to minimize damage caused by gully erosion, Pal et al. [5] developed GES maps for the Garhbeta I Community Development (C.D.) Block in West Bengal, India, by utilizing a machine learning algorithm (MLA) of boosted regression tree (BRT), bagging and the ensemble of BRT-bagging with K-fold cross validation (CV) re-sampling techniques. Amongst the various methods adopted to evaluate and predict soil erodibility, ranging from experimental to empirical to evolutionary computational approaches, attention has not been focused on the use and comparison of the performance of genetic programming, artificial neural networks and evolutionary polynomial regression in predicting the erodibility of nanocomposite treated unsaturated soils.

By and large, the soil erodibility influencers which are factors of its mechanical and physical properties affect each other significantly. There are a number of engineering soil performance factors influenced by soil erosion such as stability, durability, swell resistance, soil slope and available water holding capacity and soil texture. Collectively, these properties affect the strength carrying potential of the soil; each of these are manifested by a change in the soil's mechanical and/or physical properties induced by the stabilization material

and the erosion mechanism such as clay content, proportion of hybrid cement by weight, coefficient of uniformity and curvature [36], plasticity index and unsaturated unit weight of soil [8,34,41]. Consequently, there is a regular need to evaluate the values and non-linear and dynamic responses of these parameters induced by the incorporation of stabilizing hybrid cement (nanotextured quarry fine (NQF) and hydrated lime activated nanostructured rice husk ash (HANRHA)) to enhance its performance abilities such as resistance to erosion actions using various evolutionary techniques [36]. The present study is aimed at evaluating and predicting the erodibility of nanocomposite treated unsaturated soil using genetic programming (GP), artificial neural network (ANN) and evolutionary polynomial regression (EPR) techniques by employing the following influencing parameters: hybrid cement (HC), clay content (C), coefficients of curvature and uniformity (Cc, Cu), plasticity index (Ip), and unsaturated unit weight.

## 2. Materials and Methods

### 2.1. Materials

The reference soil was collected from the erosion site at Amuzukwu, Nigeria as shown in Figure 1. Amuzukwu is located in Umuahia North, Abia State, Nigeria, the erosion maps of which are presented in Figures 2–5. In Figure 2, the gullies spread in Abia State are identified and mapped while the areas prone to gully development are presented in Figure 3, showing the central Abia State where theAmuzukwu community is located rated as a very high erosion area. The erosion extent in the studied area is presented in Figure 4, which shows how devastating this menace has been. And in Figure 5, it has been shown that the estimated soil loss in a year is over 36 t-ha-1/yr, and this constitutes an emergency [16]. This necessitated the collection of the studied samples towards predicting a more sustainable model for the future design and monitoring of the performance of the environment. The soil was prepared by adhering to the requirements of BS1377 [42] by removing lumps and sun drying them for three days for use in the experiment.

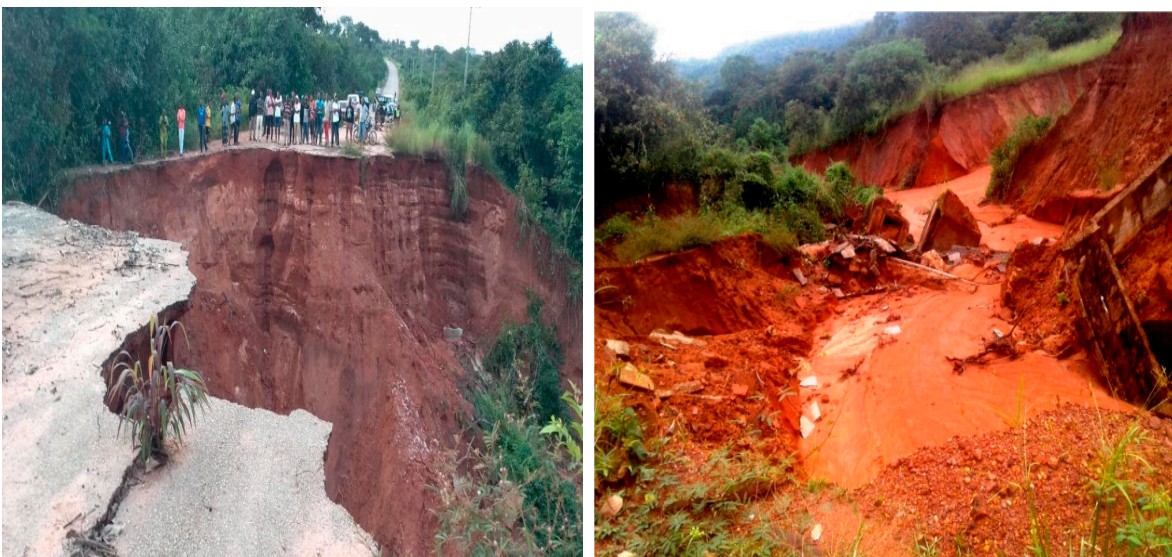

**Figure 1.** Amuzukwu, Umuahia North Local Government Area, Abia State, Nigeria erosion sites.

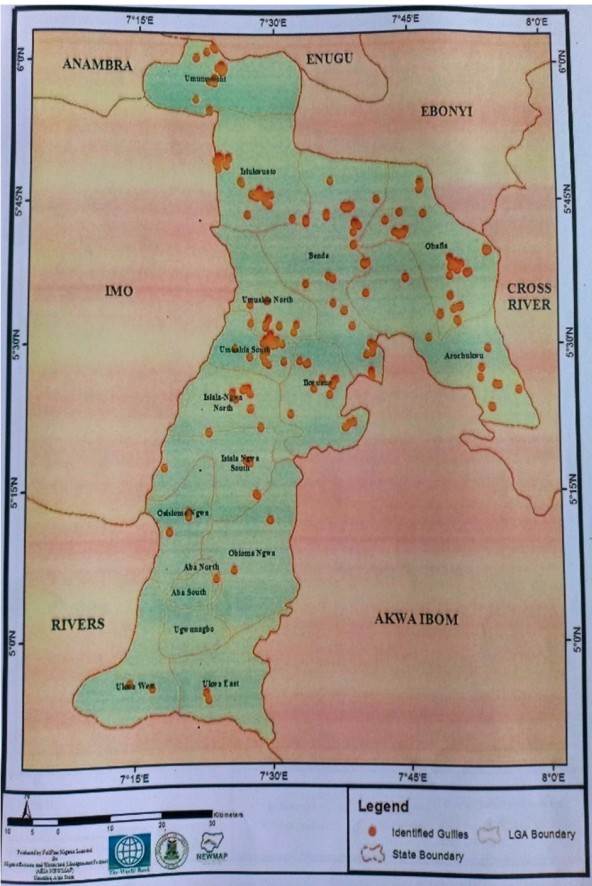

**Figure 2.** Identified Gullies in Abia State, Nigeria [16].

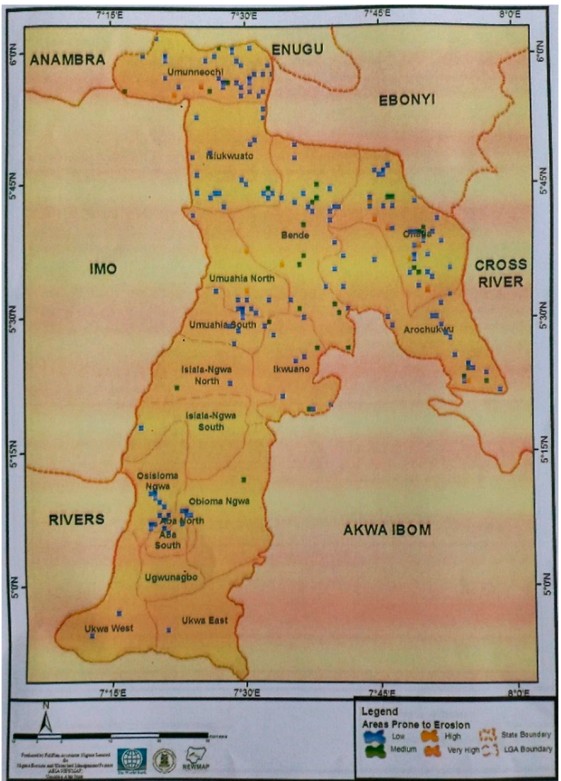

**Figure 3.** Location of Areas Prone to Erosion Between Low and Very High Intensity [16].

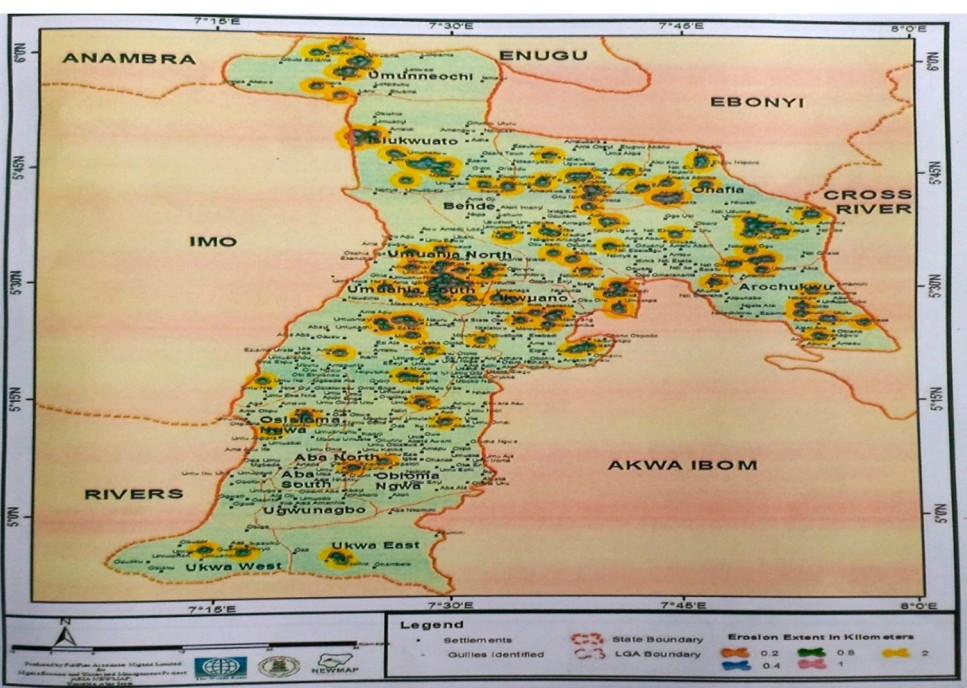

**Figure 4.** Erosion Extent in the Studied Area in Kilometers [16].

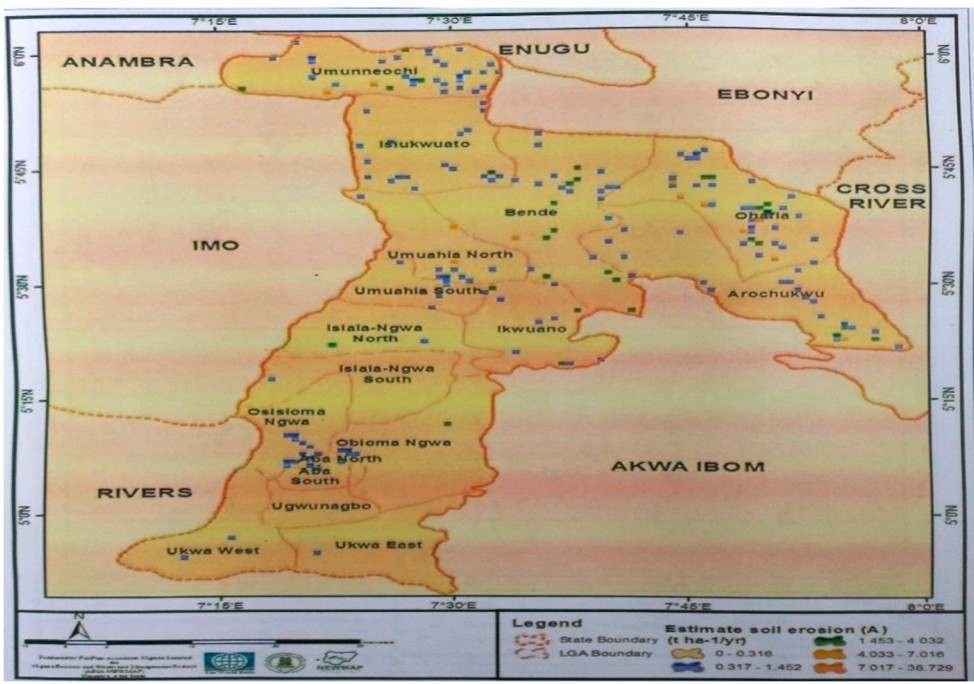

**Figure 5.** Estimated Soil Erosion in the Studied Area in t-ha-1/yr [16].

Rice husk ash (RHA) was generated by combusting rice husk (RH) collected from farm-side dumps and mills in Ebonyi State in a controlled incinerator mechanism developed by K. C. Onyelowe et al. [43], allowed to cool and was sieved through 200-nanometer sieve to produce nanotextured rice husk ash (NRHA) [44–47]. Furthermore, 5% hydrated-lime by weight of NRHA was mixed with NRHA and was left for 48 h to generate hydrated-lime activated nanotextured rice husk ash (HANRHA). Nanotextured quarry fines (NQF) were derived by passing quarry dust through a 200-nanometer sieve. Finally, HANRHA and NQF were deeply mixed for 15 min to ensure homogeneity to generated hybrid cement

(HC), which forms a highly cementitious composite with high pozzolanic performance in line with the requirements of ASTM C618 [48].

*2.2. Methods*

2.2.1. Experimental Methods

Preliminary experiments were conducted to classify the soil involving particle size analysis, Atterberg limits, compaction, specific gravity, a swelling test, etc., in accordance with laboratory conditions of BS1377 [42]. Furthermore, the HC was utilized in the proportions of 0.1% to 12% and increased at the rate of 0.1% to treat the soil, and tests were conducted in accordance with the conditions of BS1924 [49], which included clay content, coefficients of curvature and uniformity, unsaturated unit weight, Atterberg limits, and erodibility tests. Multiple datapoints were generated.

2.2.2. Collected Database; Statistical Analysis, Distribution and Correlation of Parameters

At the end of the soil improvement exercise by HC treatment and testing of the prepared specimens, several datapoints were determined on the following physical and mechanical properties as stated earlier with a functional $Er = f\ (HC, C, Cc, Cu, \gamma_{unsat}, Ip)$ used as the fundamental principle in the model prediction:

- Hybrid Cement percent by weight (HC),
- Clay content (C)
- Coefficient of curvature (Cc)
- Coefficient of uniformity (Cu)
- Unsaturated unit weight (g/cm$^3$) ($\gamma_{unsat}$)
- Plasticity index (Ip),
- Erodibility (Er)(g/s)

The measured records were divided into a training set (81 records) and a validation set (40 records). Supplementary Material Table S1 includes the complete dataset, while Tables 1 and 2 summarizes their statistical characteristics and the Pearson correlation matrix. In Table 2, Er shows the highest correlation with unsaturated unit weight behavior with the addition of HC on the soil and this proves its influence on the model. Finally, Figure 6 shows the histograms for both inputs and outputs.

**Table 1.** Statistical analysis of collected database.

| | Hybrid Cement HC | Clay Content C | Coeff. of Curvature Cc | Cu | Unsaturated Unit Weight $\gamma_{unsat}$ | Plasticity Index Ip | Erodibility (Er) |
|---|---|---|---|---|---|---|---|
| | - | - | - | - | g/cm$^3$ | - | (g/s) |
| | | | | Training set | | | |
| Max. | 0.00 | 0.23 | 0.84 | 2.05 | 1.40 | 0.14 | 0.06 |
| Min | 0.12 | 0.24 | 1.93 | 5.85 | 2.07 | 0.45 | 0.20 |
| Avg | 0.06 | 0.23 | 1.39 | 3.68 | 1.68 | 0.32 | 0.13 |
| SD | 0.03 | 0.00 | 0.28 | 1.27 | 0.18 | 0.08 | 0.03 |
| Var | 0.58 | 0.01 | 0.20 | 0.34 | 0.11 | 0.27 | 0.20 |
| | | | | Validation set | | | |
| Max. | 0.00 | 0.23 | 0.88 | 2.06 | 1.42 | 0.14 | 0.06 |
| Min | 0.12 | 0.24 | 1.96 | 5.86 | 2.07 | 0.45 | 0.19 |
| Avg | 0.07 | 0.24 | 1.46 | 4.03 | 1.74 | 0.29 | 0.12 |
| SD | 0.04 | 0.00 | 0.33 | 1.38 | 0.22 | 0.10 | 0.04 |
| Var | 0.58 | 0.01 | 0.22 | 0.34 | 0.13 | 0.35 | 0.29 |

**Table 2.** Pearson correlation matrix of the input and output parameters.

|  | **Hc** | **C** | **Cc** | **Cu** | $\gamma_{unsat}$ | **Ip** | **Er** |
|---|---|---|---|---|---|---|---|
| Hc | 1 | | | | | | |
| C | 0.99644 | 1 | | | | | |
| Cc | 0.99461 | 0.984554 | 1 | | | | |
| Cu | 0.982765 | 0.983685 | 0.965768 | 1 | | | |
| $\gamma_{unsat}$ | 0.989468 | 0.993766 | 0.981575 | 0.970896 | 1 | | |
| Ip | −0.99652 | −0.9982 | −0.98578 | −0.98582 | −0.99131 | 1 | |
| Er | −0.94097 | −0.93884 | −0.95226 | −0.89402 | −0.96073 | 0.937162 | 1 |

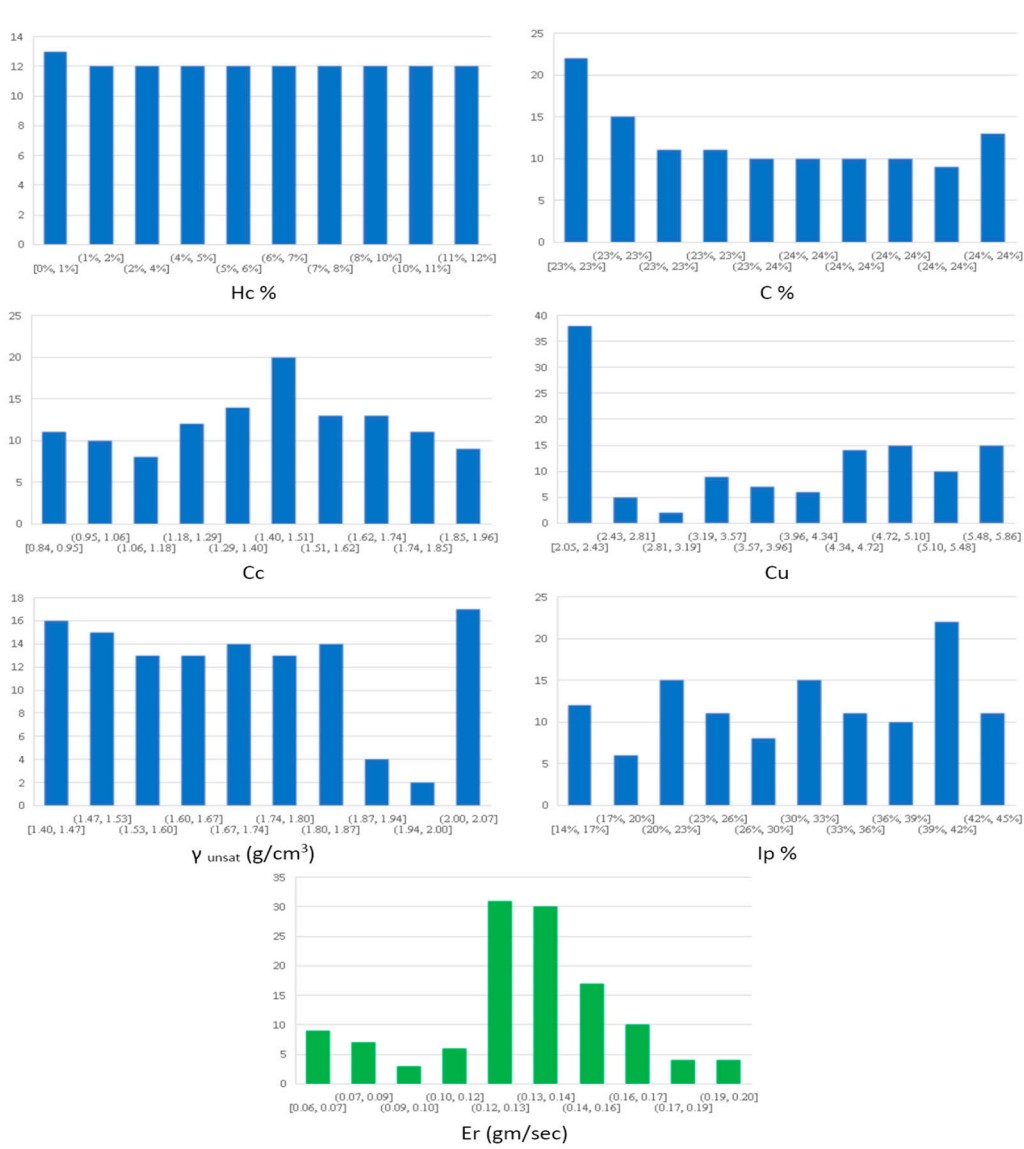

**Figure 6.** Distribution histograms for inputs (in blue) and outputs (in green).

### 2.2.3. Research Model Program

Three different artificial intelligent (AI) techniques were used to predict the shear strength parameters of the tested soil samples. These techniques are genetic programming (GP), artificial neural network (ANN) and polynomial linear regression optimized using the genetic algorithm, which is known as evolutionary polynomial regression (EPR). All of the three developed models were used to predict the values of erodibility (Er) using the measured hybrid cement percent by weight (HC), clay content (C), coefficient of curvature

(Cc), coefficient of uniformity (Cu), unsaturated unit weight ($\gamma_{unsat}$) and plasticity index (Ip). Each model of the three developed models was based on a different approach (evolutionary approach for GP, mimicking biological neurons for ANN and optimized mathematical regression technique for EPR). However, for all developed models, prediction accuracy was evaluated in terms of sum of squared errors (SSE).

The following section discusses the results of each model. The accuracies of developed models were evaluated by comparing the (SSE) between predicted and calculated erodibility (Er) values. The results of all developed models are summarized in Table 3.

**Table 3.** Preliminary Properties of the Reference Soil at 60% Saturation.

| Property | % Passing 0.075 mm | NMC | LL | PL | PI | SP | SG | AASHTO | MDD | OMC |
|---|---|---|---|---|---|---|---|---|---|---|
| Value | 44 | 12 | 62 | 21 | 39 | 25 | 1.20 | A-7-6 | 1.15 | 17 |
| Unit | % | % | % | % | % | % | - | - | g/cm$^3$ | % |

## 3. Discussion of Results

### 3.1. Materials' Properties

Table 3 contains the preliminary properties of the reference soil, which showed high plasticity of over 17%, high swelling potential (25%), low dry density of 1.15 g/cm$^3$ obtained at the optimum moisture of 17%, low specific gravity of 1.2 and classified as A-7-6 group according totheAASHTO classification system. These properties show that the soil lacked the fundamental requirement to be used as a foundation material and more so in a hydraulically bound environment. Table 4 shows the oxide composition of the soil, NRHA, NQF and HC. This shows the high pozzolanic content of the composites (NRHA and NQF) with which the supplementary cementitious binds with improved pozzolanity (97.15%), which meets the requirements of ASTM C618 [49] and BS 8615-1 [50], compared to the NQF and NRHA. This is due to the blending of two relatively pozzolanic nanotextured materials which triggered carbonation, polymerization, further activation and flocculation reactions. Tables 5 and 6 and Figures 7 and 8 show the absorbance and wavelength micrographic variation for the NRHAand NQF using an ultraviolet visual spectrophotometer at 25 °C. These show the nucleating capacity of the materials due to the increased surface area through nanosization. Though the NQF showed a higher absorbance size of 1.217 nm compared to an NRHA of 1.12 nm, it was achieved at a relatively higher wavelength of 800 nm. This implies that NRHA, even at a lower wavelength of 650 nm, can blend and form a homogenous mixture as an SCM in soil treatment.

**Table 4.** Chemical oxide composition of the additive materials.

| Materials | Oxides' Composition (Content by Weight,%) | | | | | | | | | | | | |
|---|---|---|---|---|---|---|---|---|---|---|---|---|---|
| | SiO$_2$ | Al$_2$O$_3$ | CaO | Fe$_2$O$_3$ | MgO | K$_2$O | Na$_2$O | TiO$_2$ | LOI | P$_2$O$_5$ | SO$_3$ | *IR | Free CaO |
| Clay Soil | 8.45 | 13.09 | 2.30 | 10.66 | 4.89 | 17.00 | 37.33 | 1.17 | - | 5.11 | - | - | - |
| NRHA | 57.48 | 22.72 | 4.56 | 3.77 | 4.65 | 2.76 | 0.01 | 3.17 | 0.88 | - | - | - | - |
| NQF | 62.48 | 18.72 | 4.83 | 6.54 | 2.56 | 3.18 | - | 0.29 | 1.01 | - | - | - | - |
| HC | 66.5 | 27.8 | 1.3 | 2.85 | 1.5 | 0.03 | - | 0.02 | - | - | - | - | - |

*IR is insoluble residue; LOI is loss on ignition.

In previous research results presented by Onyelowe et al. [51], it was observed in Figure 9 that "RHA exhibited gel-like porous-valve structure at a magnification level of 100 μm. Additionally, the presence of bigger voids could be observed in the SEM micrograph, which illustrates the light-weight structure and porous structure of the agricultural waste". It can be seen from Figure 10 that the scanning electron micrograph (10 μm) of quarry dust (QD) with ×1500 magnification and at 20 kV depicts the microstructural arrangement of the particles, where the bright portion of the image represents a lesser dense structure which is reflected from the smaller value of the specific gravity of the QD.

Additionally, the traces of smaller fragments are an indication of the relatively larger and finer surface area of the QD particles.

**Table 5.** Absorbance and wavelength variation for the NRHA using an ultraviolet visual spectrophotometer at 25 °C [45].

| Wavelength (nm) | Absorbance (nm) |
| --- | --- |
| 0 | 1.116 |
| 200 | 1.115 |
| 250 | 1.115 |
| 300 | 1.106 |
| 350 | 1.103 |
| 400 | 1.106 |
| 450 | 1.105 |
| 500 | 1.094 |
| 550 | 1.066 |
| 600 | 1.045 |
| 650 | 1.120 |
| 700 | 1.003 |
| 750 | 1.062 |
| 800 | 1.045 |
| 850 | 1.031 |
| 900 | 1.045 |
| 950 | 1.070 |
| 1000 | 1.091 |

**Table 6.** Absorbance and wavelength variation for the NQF using an ultraviolet visual spectrophotometer at 25 °C [45].

| Wavelength (nm) | Absorbance (nm) |
| --- | --- |
| 0 | 1.216 |
| 200 | 1.007 |
| 250 | 0.015 |
| 300 | 1.036 |
| 350 | 0.903 |
| 400 | 0.416 |
| 450 | 1.005 |
| 500 | 1.154 |
| 550 | 1.216 |
| 600 | 0.075 |
| 650 | 0.102 |
| 700 | 1.003 |
| 750 | 1.212 |
| 800 | 1.217 |
| 850 | 0.931 |
| 900 | 0.245 |
| 950 | 1.070 |
| 1000 | 0.191 |

*3.2. Intelligent Predictionsof Erodibility (Er)*

3.2.1. Model (1)—Using (GP) Technique

The developed GP model started with the one level of complexity and settled at four levels of complexity. The population size, survivor size and number of generations were 100,000, 30,000 and 200 respectively. Equation (1) presents the output formula for (Er), generated from the four levels complexity trail, which agrees with previous research [51] to

present the highest model performance compared to one and third levels of complexity. The average error % of the total set for (Er) is (3.9%), while the ($R^2$) value is (0.974).

$$Er = \frac{2(3HC + 2\gamma_{unsat})}{(120\,Cc\,*\,Ip) + Ip(3HC + 2\gamma_{unsat})(Cc^{1.2Cu\,+1} - 30\,Ip)} \tag{1}$$

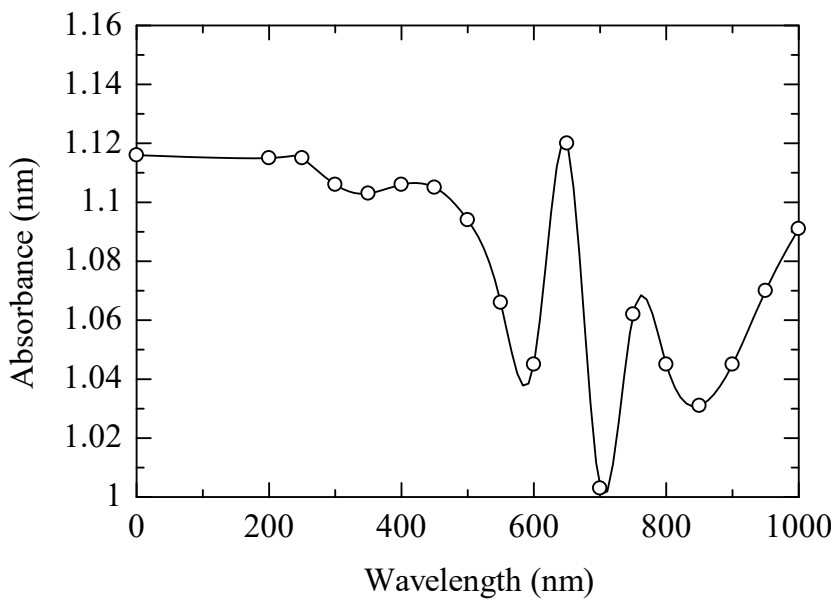

**Figure 7.** Absorbance and wavelength variation for the NRHA using an ultraviolet visual spectrophotometer at 25 °C [45].

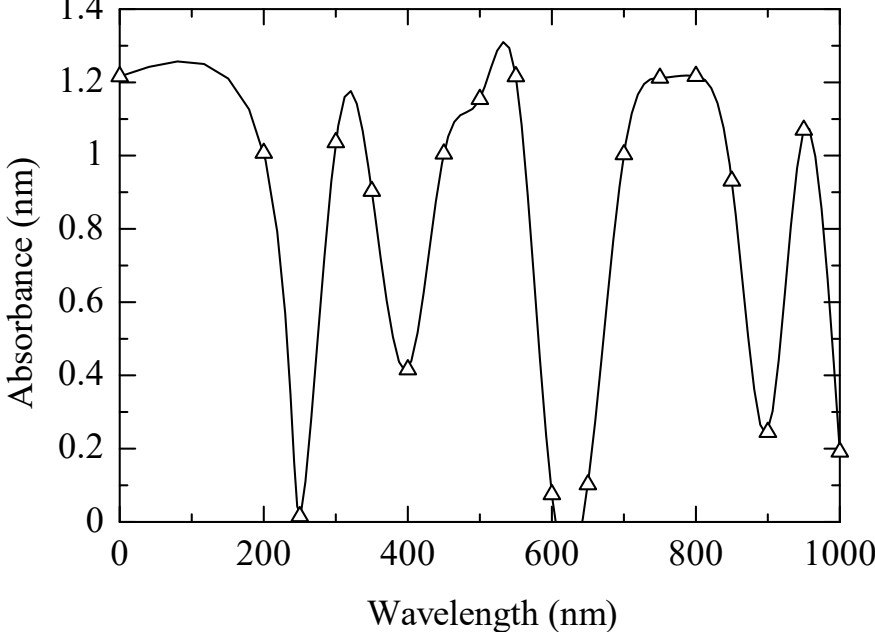

**Figure 8.** Absorbance and wavelength variation for the NQF using an ultraviolet visual spectrophotometer at 25 °C [45].

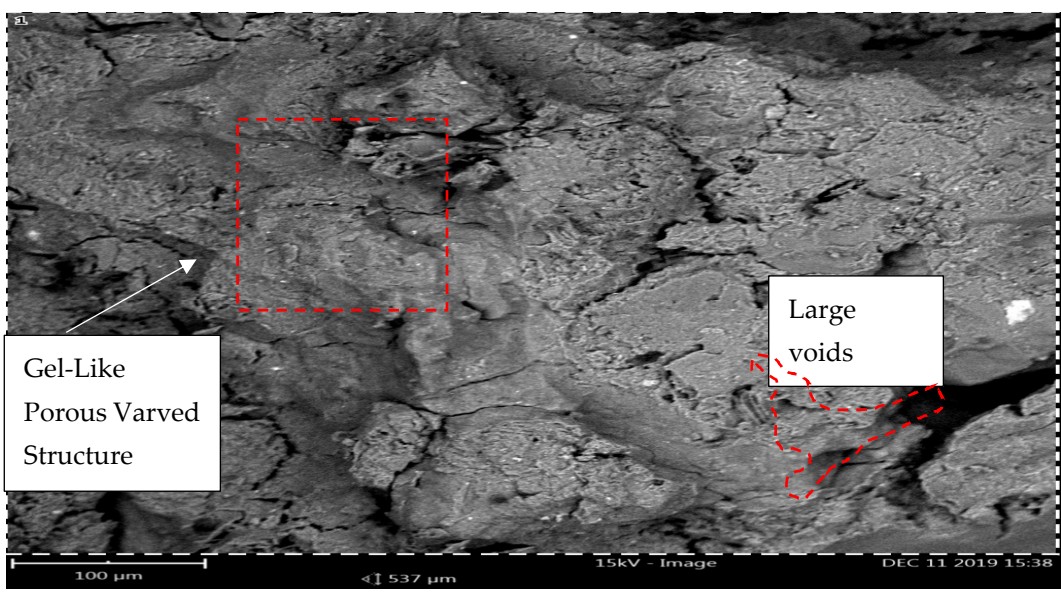

**Figure 9.** Morphology of silica-rich RHA.

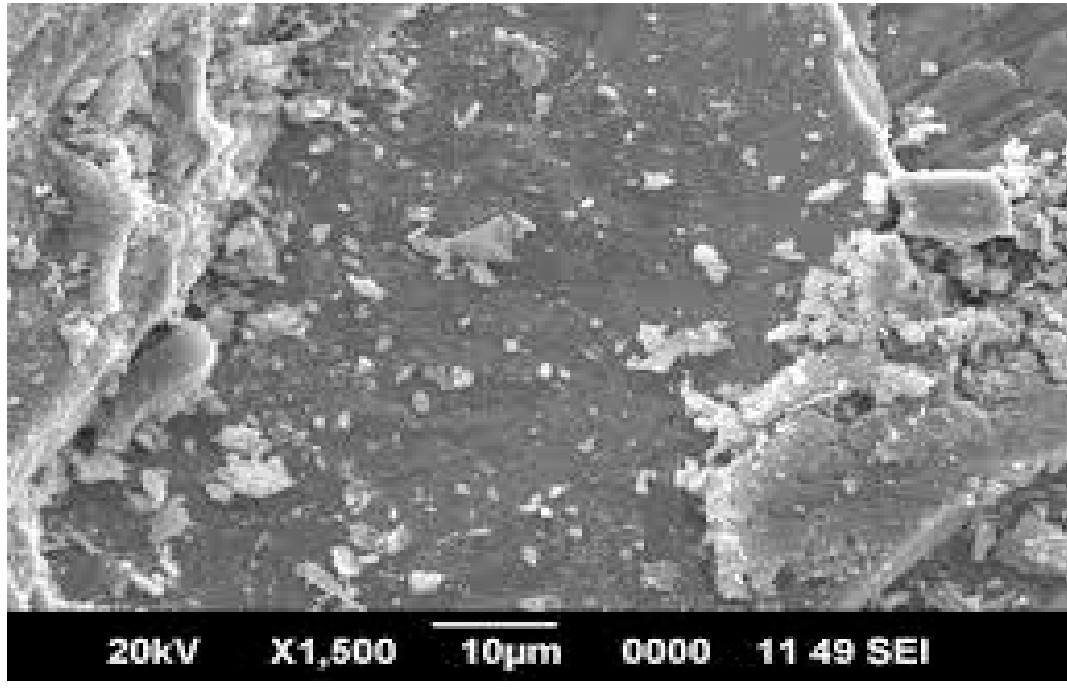

**Figure 10.** Morphology of quarry dust (QD).

3.2.2. Model (2)—Using (ANN) Technique

A back propagation ANN with one hidden layer and (HyperTanh) activation function was used to predict the same erodibility (Er) values. The used network layout and its connation weights are illustrated in Figure 11a. Since the used ANN has a nonlinear activation function, the first procedure that gave rise to the performance indices of 4.9% and 0.954 for average error (SSE) and coefficient of determination($R^2$), respectively, utilized the "Tahn" activation function with standardized operation ranges from −1.0 to 1.0 and presented a network layout with its connation weights as shown in Figure 11a, while the "sigmoid" activation function with an operation range between 0.0 to 1, was applied to generate both the sigmoid network layout with its connation weights in Figure 11b and the complex-form equivalent equations represented in Equation (2), which is supported via the

substitution of parameters by Equations (3)–(5). This sigmoid activation function model operation produced an average error % of the total dataset for this network as (3.2%), and the coefficient of determination ($R^2$) value as (0.979). This shows the superiority of the sigmoid activation function, though complicated with functions y = 1/(1+Exp(−x)) applied to the ANN prediction operations with better performance. Accordingly, for positive input values, "sigmoid" is fully functioning while "Tahn" is only half-functioned and this contributes to its (sigmoid) performance. The relation between calculated and predicted values is shown in Figure 12.

$$Er = 0.06 + \frac{0.14}{1 + e^{-Y1}}. \tag{2}$$

$$Y1 = 0.12 - \frac{6.533}{1 + e^{-X1}} + \frac{4.128}{1 + e^{-X2}} \tag{3}$$

$$
\begin{aligned}
X1 &= -4.415 - 0.672\,Hc' - 0.249\,C' + 2.224\,Cc' - 1.762\,Cu' + 4.791\gamma'_{unsat} - 3.012\,Ip' \\
X2 &= -0.702 - 2.577\,Hc' - 1.056\,C' - 3.961\,Cc' + 0.139\,Cu' - 1.905\gamma'_{unsat} + 0.679\,Ip'
\end{aligned} \tag{4}
$$

$$Hc' = \frac{Hc}{0.12};\ C' = \frac{C-0.23}{0.01};\ Cc' = \frac{Cc-0.84}{1.12};\ Cu' = \frac{Cu-2.05}{3.81};\ \gamma'_{unsat} = \frac{\gamma_{unsat}-1.4}{0.67};\ Ip'$$
$$= \frac{Ip-0.14}{0.31} \tag{5}$$

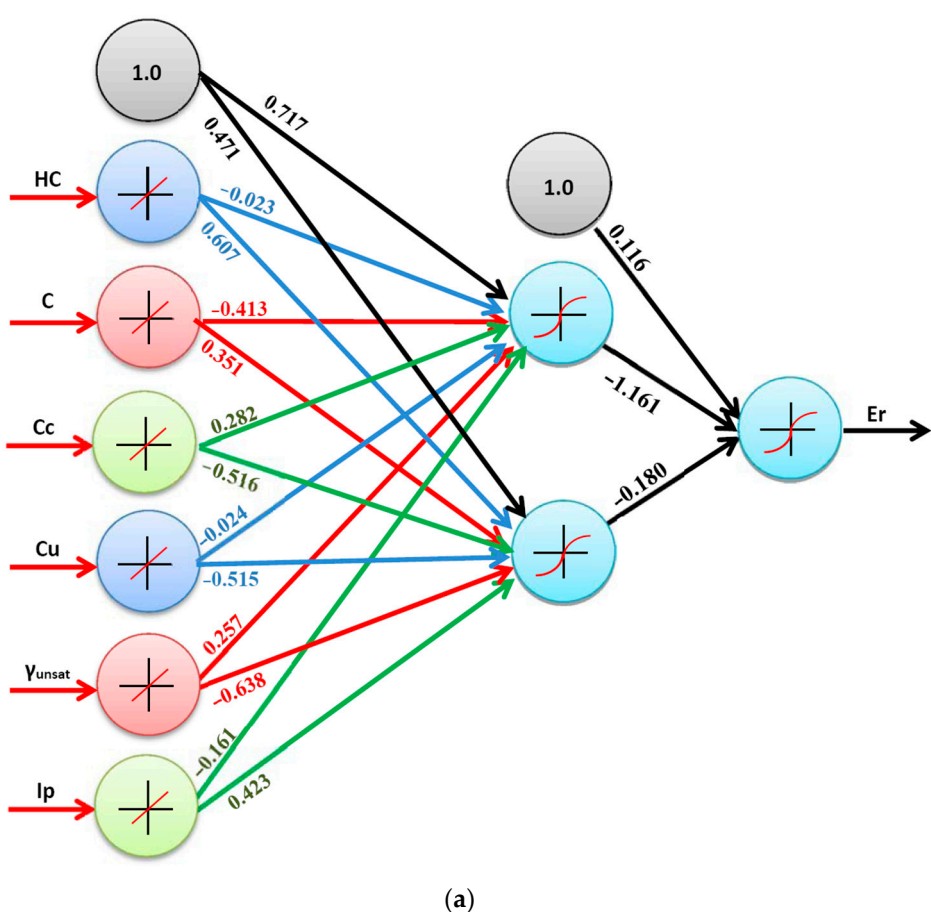

(a)

**Figure 11.** *Cont.*

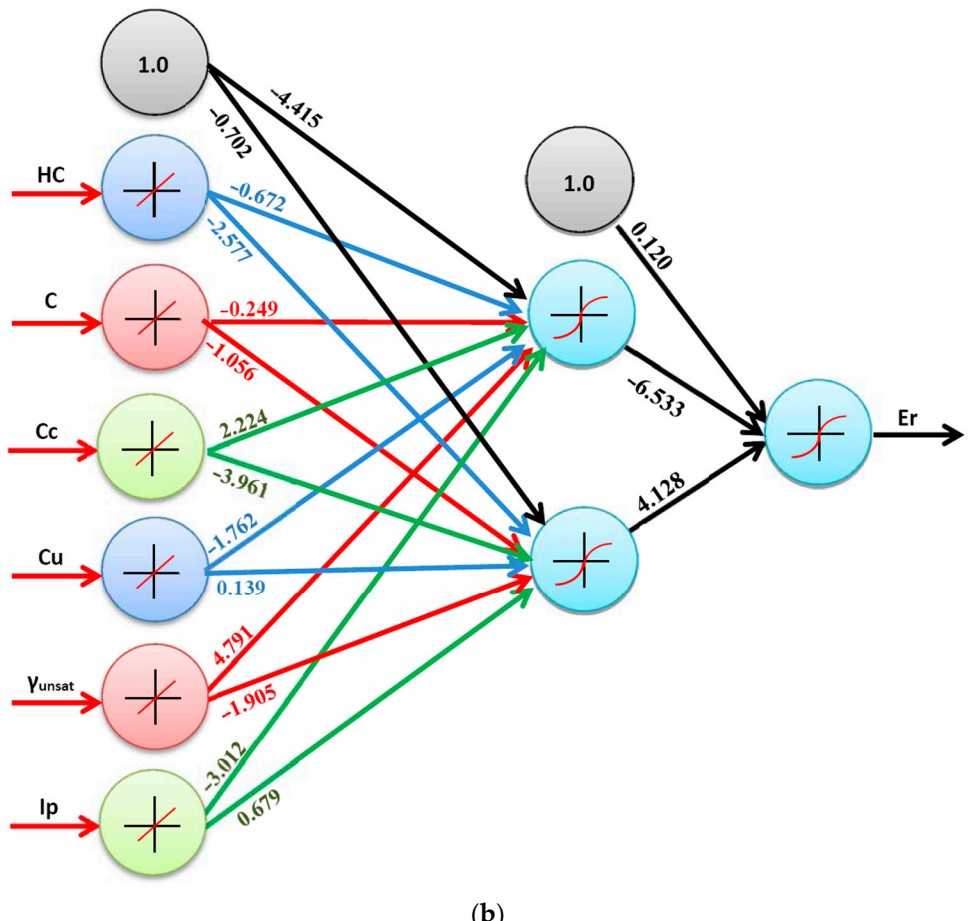

(**b**)

**Figure 11.** (**a**) ANN-Hyper-Tanh network layout for the developed ANN and its connection weights. (**b**) ANN-Sigmoid network layout for the developed ANN and its connection weights.

### 3.2.3. Model (3)—Using (EPR) Technique

Finally, the developed EPR model was limited to the quadratic level; for six inputs, there are 43 possible terms ($\sum_{i=1}^{i=6} X_i + \sum_{i=1}^{i=6} \sum_{j=1}^{j=6} X_i.X_j + C$). A GA optimized regression technique was applied on these 43 terms to select the most effective 11 terms to predict the values of the erodibility (Er). The output is illustrated in Equation (6) and its fitness is shown in Figure 12. The average error% and ($R^2$) values were improved to (1.6%)–(0.996) for the total datasets, respectively. Finally, Table 7 shows the summary of the performance of the developed models. Finally, the line of best fits between measured and modeled values is presented in Figure 12.

$$Er = 8.236\,Ip^2 \;+ 68.826\,HC^2 + 0.539\,Cc^2 - 8.505\,HC \cdot Cc - 4.61\,C \cdot Cc - 6.173\,C \cdot \gamma_{unsat} \atop + 6.704\gamma_{unsat} \cdot Ip + 23.566\,C - 16.43\,Ip - 1.233 \tag{6}$$

**Table 7.** Performance accuracies of developed models.

| Intelligent Technique | Developed Equation | Error% | $R^2$ |
|---|---|---|---|
| GP (4 levels of complexity) | Equation (1) | 3.9 | 0.974 |
| ANN (tahn) | Figure 11a | 4.9 | 0.954 |
| ANN (sigmoid) | Figure 11b; Equation (2) | 3.2 | 0.979 |
| EPR (GA optimized) | Equation (2) | 1.6 | 0.996 |

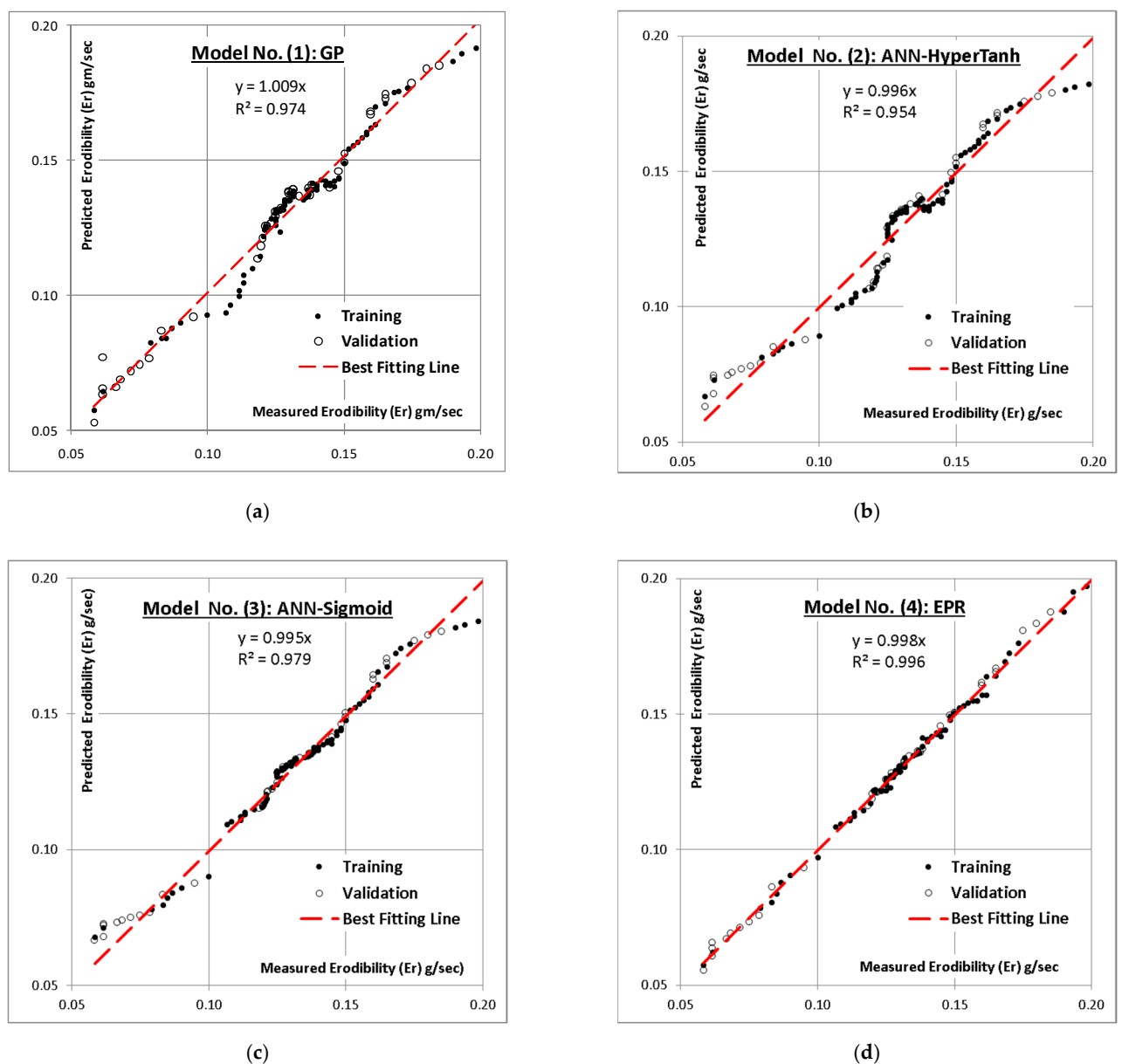

**Figure 12.** Relation between predicted and calculated (Er); (**a**) GP, (**b**) ANN-HyperTanh, (**c**) ANN-Sigmoid, (**d**) EPR values using the developed models.

## 4. Conclusions

This research presents three models using three (AI) techniques (GP, ANN and EPR) to predict the erodibility (Er) values using the measured hybrid cement percent by weight (HC), clay content (C), coefficient of curvature (Cc), coefficient of uniformity (Cu), unsaturated unit weight ($\gamma_{unsat}$) and plasticity index (Ip). The results of comparing the accuracies of the developed models can be summed in the following points:

- The prediction accuracies of the ANN (sigmoid), ANN (Tahn) and GP models are close (97.9%, 95.4% and 97.4%), which gives an advantage to the ANN (sigmoid activation function) model because its output is a closed form equation and could be applied either manually or implemented in software. On the other hand, the prediction accuracy of the EPR model is better than all of them (99.6%; 1.6%); in addition, its output is a closed form equation, and that makes it the optimum model.

- The outputs of both the GP and the ANN models indicated that (Er) values were mainly governed by (Ip) & ($\gamma_{unsat}$), while (Cu), (Cc) and (HC) have a secondary impact, and (C) has a negligible impact on (Er) values.
- The erodibility value (Er) decreased with decreasing (Ip) values and with increasing ($\gamma_{unsat}$) values.
- The GA optimized technique successfully reduced the 43 terms of conventional polynomial linear regression (PLR)quadratic formula to only 11 terms without significant impact on its accuracy.
- Like any other regression technique, the generated formulas are valid within the considered range of parameter values; beyond this range, the prediction accuracy should be verified.

**Supplementary Materials:** The following supporting information can be downloaded at: https://www.mdpi.com/article/10.3390/su14127403/s1, Table S1: The used database.

**Author Contributions:** Conceptualization, K.C.O.; Data curation, A.M.E.; Formal analysis, K.C.O. and H.N.O.; Methodology, K.C.O.; Project administration, K.C.O.; Resources, I.O. and A.A.F.; Software, A.M.E.; Supervision, K.C.O. and M.E.O.; Validation, A.M.E.; Writing—original draft, K.C.O., A.M.E., U.E. and L.I.N.; Writing—review & editing, K.C.O., U.E., M.E.O. and A.A.F. All authors have read and agreed to the published version of the manuscript.

**Funding:** The authors received no funding for this research work.

**Data Availability Statement:** The data supporting the findings of this research work has been presented in the manuscript.

**Acknowledgments:** The authors are grateful to Engr. IzuchukwuOnwughara of the Abia State NEWMAP, for his magnanimity in releasing the erosion maps which have added quality and content to the present research work.

**Conflicts of Interest:** The authors declare that they have no conflict of interest.

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
