# Peer review of "Erodibility of Nanocomposite-Improved Unsaturated Soil Using Genetic Programming, Artificial Neural Networks, and Evolutionary Polynomial Regression Techniques"

_sustainability, doi:10.3390/su14127403_

Round 1
Reviewer 1 Report
The paper used AI technology to predict soil erosion-related problems. The method is sound. The prediction results are satisfactory. However, overall, the paper still needs to be revised. Please consider the following:
- Reference style should be revised to meet the requirement of the Journal.
- English style is the primary problem. Currently, language style can be confusing to readers.
- The quality of some figures is poor, such as Fig. 1-Fig. 10. Please replace those with clearer ones.
- The introduction requires revising. For example, the impact of soil erosion on infrastructure is also significant as this can cause extra stress and strain on the facility, leading to safety issues. The following papers may help authors improve the Introduction as some background information may be enlightening to the author:
https://doi.org/10.1016/j.psep.2019.02.028;
http://dx.doi.org/10.1016/j.scitotenv.2022.154725;
https://doi.org/10.1016/j.compag.2021.105990.
- Discussions need to increase the broad applicability of the model.
Author Response
- Reference style should be revised to meet the requirement of the Journal.
The references have been revised to meet journal requirements
- English style is the primary problem. Currently, language style can be confusing to readers.
The authors have edited the manuscript
- The quality of some figures is poor, such as Fig. 1-Fig. 10. Please replace those with clearer ones.
The resolutions of the figures have been improved beyond 300 pixels
- The introduction requires revising. For example, the impact of soil erosion on infrastructure is also significant as this can cause extra stress and strain on the facility, leading to safety issues. The following papers may help authors improve the Introduction as some background information may be enlightening to the author:
The authors have cited sufficient and relevant published articles in this research work.
https://doi.org/10.1016/j.psep.2019.02.028;
http://dx.doi.org/10.1016/j.scitotenv.2022.154725;
https://doi.org/10.1016/j.compag.2021.105990.
- Discussions need to increase the broad applicability of the model.
This section contains sufficient information to allow readers and researchers understand the findings of the research as to apply it in future research activities and in the field.
Reviewer 2 Report
Thank you for the opportunity to review this interesting article. The manuscript is well written but I have a few comments on it which should be taken into account before publication.
Line 47 – please use SI units in the manuscript
The quality of figures 1-5 is not appropriate for publishing; figures have to be correctly sized and of good quality. Either, use the original digital material or make new, good-quality scans.
Line 311 - In units of area and volume always use superscript.
Figures 7 and 8 - must be replaced by figures with good text sharpness.
Figure 9 – the text in frames is cut off
Figures 10 - must be replaced by figures with good sharpness.
I consider the presentation of the results and the discussion to be appropriate. The conclusions are clearly written.
Author Response
Line 47 – please use SI units in the manuscript
The manuscript has been revised and SI units have been used
The quality of figures 1-5 is not appropriate for publishing; figures have to be correctly sized and of good quality. Either, use the original digital material or make new, good-quality scans.
The figures resolution has been revised for clarity
Line 311 - In units of area and volume always use superscript.
scripts have been applied accordingly where necessary all through the manuscript
Figures 7 and 8 - must be replaced by figures with good text sharpness.
The resolution of the figures have been improved for clearer sharpness
Figure 9 – the text in frames is cut off
This has also been revised. Thank you
Figures 10 - must be replaced by figures with good sharpness.
The sharpness of the figure has been improved
I consider the presentation of the results and the discussion to be appropriate.
Thank you.
The conclusions are clearly written.
Thank you
Reviewer 3 Report
The article concerns research on erodibility of nanocomposite-improved unsaturated soil using genetic programming, artificial neural networks, and evolutionary polynomial regression techniques. The article should be corrected before publication, because it is prepared carelessly.
From the scientific point of view, the article presents the average scientific level.
The paper should be improved namely in some identified aspects:
- the authors should have some extra effort to make the description more easily understandable, more clear to all the readers and more simple to read as the subject seems a kind of too “complicated”.
- figures ARE NOT CLEAER !!! There are copies of the photos taken carelessly and the descriptions are illegible. This is not acceptable.
- although the English is OK, the text needs to be simplified.
- the achievements presented in the article should be compared more closely with examples from other countries.
Author Response
- the authors should have some extra effort to make the description more easily understandable, more clear to all the readers and more simple to read as the subject seems a kind of too “complicated”.
- This has been revised with the English language editing, the context of the work is now simple and clearer to read
- figures ARE NOT CLEAER !!! There are copies of the photos taken carelessly and the descriptions are illegible. This is not acceptable.
- The figures were not taken carelessly as pictures. they were prepared as an official mapping system for the government of the state with the highest number of erosion sites in Nigeria. The resolution of the pictures has been improved as a consequence of this review
- although the English is OK, the text needs to be simplified.
- The simplification exercise has been conducted, thank you.
- the achievements presented in the article should be compared more closely with examples from other countries.
- Erosion problem is a global environmental phenomenon and it is related to wherever this menace exists. Meanwhile intelligent models have been proposed for application due to slope stabilization
Round 2
Reviewer 1 Report
In the revision version, I have not seen a manuscript with traces of revision. Please provide it.
Author Response
I have attached the manuscript with tracked revisions.
Authors' Response:
Comment #1: Reference style should be revised to meet the requirement of the Journal.
Response: The references have been revised to meet journal requirements
Comment #2: English style is the primary problem. Currently, language style can be confusing to readers.
Response: The authors have edited the manuscript
Comment #3: The quality of some figures is poor, such as Fig. 1-Fig. 10. Please replace those with clearer ones.
Response: The resolutions of the figures have been improved beyond 300 pixels
Comment #4: The introduction requires revising. For example, the impact of soil erosion on infrastructure is also significant as this can cause extra stress and strain on the facility, leading to safety issues. The following papers may help authors improve the Introduction as some background information may be enlightening to the author:
Response: The authors have cited sufficient and relevant published articles in this research work.
https://doi.org/10.1016/j.psep.2019.02.028;
http://dx.doi.org/10.1016/j.scitotenv.2022.154725;
https://doi.org/10.1016/j.compag.2021.105990.
Comment #5: Discussions need to increase the broad applicability of the model.
Response: This section contains sufficient information to allow readers and researchers understand the findings of the research as to apply it in future research activities and in the field.

Reviewer 3 Report
The authors corrected the article in relation to the previous version, however, a few points were not corrected - the quality of figures 2, 3, 4, 5, 7, 8 leaves many doubts. I have no objections to the scientific quality of the article.
Author Response
sustainability-1684287
Reviewer 3
The authors corrected the article in relation to the previous version, however, a few points were not corrected - the quality of figures 2, 3, 4, 5, 7, 8 leaves many doubts. I have no objections to the scientific quality of the article.
The figures resolutions have been improved further and as can be seen in the manuscript, their DPI is now higher and clearer. Thank you for your time

Round 3
Reviewer 1 Report
Well done.
